# Is the Concurrent Use of Sorafenib and External Radiotherapy Feasible for Advanced Hepatocellular Carcinoma? A Meta-Analysis

**DOI:** 10.3390/cancers13122912

**Published:** 2021-06-10

**Authors:** Chai Hong Rim, Sunmin Park, In-Soo Shin, Won Sup Yoon

**Affiliations:** 1Department of Radiation Oncology, Ansan Hospital, Korea University Medical College, Ansan 15355, Korea; sunmini815@gmail.com (S.P.); irionyws@korea.ac.kr (W.S.Y.); 2Graduate School of Education, Dongguk University, Seoul 04620, Korea; 9065031@hanmail.net

**Keywords:** sorafenib, external beam radiation therapy, concurrent, combined, hepatocellular carcinoma, toxicity

## Abstract

**Simple Summary:**

Concurrent chemotherapy and external radiation is a commonly used method for cancer treatment. However, concurrent application of sorafenib and external radiotherapy has not been commonly used in clinical practice due to the possible risk of excessive complication. The results of this meta-analysis suggest that concurrent treatment might be a feasible option, and treatment targeting metastatic lesion or vessel involvement is particularly recommended.

**Abstract:**

We evaluate the feasibility of a concurrent application of sorafenib and external beam radiation therapy (EBRT) for advanced hepatocellular carcinoma (HCC). PubMed, Embase, Medline, and Cochrane Library were searched up to 9 April 2021. The primary endpoint was grade ≥3 complications, and the secondary endpoint was overall survival (OS). Subgroup analyses were performed for studies with the EBRT targets, intrahepatic vs. non-intrahepatic lesions (e.g., extrahepatic metastases or malignant vessel involvement only). Eleven studies involving 512 patients were included in this meta-analysis. Pooled rates of gastrointestinal, hepatologic, hematologic, and dermatologic grade ≥3 toxicities were 8.1% (95% confidence interval (CI): 4.8–13.5, I^2^ = ~0%), 12.9% (95% CI: 7.1–22.1, I^2^ = 22.4%), 9.1% (95% CI: 3.8–20.3, I^2^ = 51.3%), and 6.8% (95% CI: 3.8–11.7, I^2^ = ~0%), respectively. Pooled grade ≥3 hepatologic and hematologic toxicity rates were lower in studies targeting non-intrahepatic lesions than those targeting intrahepatic lesions (hepatologic: 3.3% vs. 17.1%, *p =* 0.041; hematologic: 3.3% vs. 16.0%, *p =* 0.078). Gastrointestinal and dermatologic grade ≥3 complications were not significantly different between the subgroups. Regarding OS, concurrent treatment was more beneficial than non-concurrent treatment (odds ratio: 3.3, 95% CI: 1.3–8.59, *p =* 0.015). One study reported a case of lethal toxicity due to tumor rupture and gastrointestinal bleeding. Concurrent treatment can be considered and applied to target metastatic lesions or local vessel involvement. Intrahepatic lesions should be treated cautiously by considering the target size and hepatic reserve.

## 1. Introduction

Hepatocellular carcinoma (HCC) has a tendency to rapidly change the tumor microenvironment and metabolic profile, resulting in tumorigenic and proliferation properties [1]. Although surveillance programs have been conducted in many countries, many patients are still diagnosed with advanced disease. Adherence to surveillance is less than 50% [2] and updated modalities could not be applied to all populations at high risk, although sonography still has a major role [3,4,5]. From a wider perspective, approximately 70% of patients were unresectable at diagnosis in China [6]; one-third of patients were found to have Barcelona Clinic of Liver Cancer (BCLC) C or higher disease in Korean National Database data [7].

External beam radiation therapy (EBRT) and chemotherapy have been used in combination to treat diverse cancers. The basic principle of this treatment includes spatial cooperation, where radiation targets macroscopic diseases, and the systemic agent treats latent microscopic disease. In addition, chemotherapy increases the effectiveness of EBRT by inducing cell cycle arrest and interfering with DNA damage repair [8,9]. Sorafenib has been established as a systemic agent that can increase the survival of patients with advanced HCC [10]. EBRT has also recently been shown to have significant effects on HCC with major vessel involvement or metastases [11,12,13]. Preclinical evidence regarding the synergistic effect of sorafenib and EBRT has been reported [14,15]. Nevertheless, few clinical studies have examined the concurrent use of sorafenib and radiotherapy. This scarcity is due to concerns about excessive toxicities [16,17,18].

Brade et al. [16] reported a serious hepatotoxicity rate of approximately 20%, which highlights the risk of concomitant therapy with EBRT and sorafenib. However, several researchers [19,20,21] have reported their experience with such combination therapy, and the degree of serious toxicities was not significantly different from that reported in the previous relevant literature (e.g., sequential treatment). In addition, as modern radiation therapy (RT) modalities such as intensity-modulated radiotherapy (IMRT) and stereotactic body radiotherapy (SBRT) become common, the targets of EBRT are becoming smaller, and consequently, the risk of toxicity caused by RT has been decreasing [22,23].

Although the concurrent treatment of sorafenib and EBRT can have synergistic benefits, clinical application is rare due to the risk of toxicity reported by a small number of previous studies. Therefore, the risks and effectiveness of concurrent applications are mostly unknown. This meta-analysis aimed to compile published data to gain a better understanding of the feasibility of using a combined treatment in advanced HCC.

## 2. Methods

### 2.1. Searching Process

We adhered to the PRISMA guidelines [24] to conduct the present systematic review and meta-analysis as well as we could, and referred to the Cochrane handbook version 6.2 for methodological direction [25]. The present study was designed to answer the following clinical PICO question: “Is the concurrent application of sorafenib and EBRT a feasible option (as compared to sequential or non-combined modalities) for patients with advanced HCC?” The studies that met the following inclusion criteria were included: (1) clinical trials involving the application of concurrent sorafenib and EBRT; (2) those including at least five patients with HCC who underwent such concurrent treatment; and (3) those with data regarding primary and/or secondary endpoints. We searched databases such as PubMed, Embase, Medline, and Cochrane Library, as recommended by the Cochrane handbook [26], for literature until 9 April 2021. The search term was designed to identify studies related to combined or concurrent use of sorafenib and EBRT for liver cancers, and the detailed search strategy is described in Appendix A. Language restrictions were not included. Chinese articles were translated by a professional Chinese–English translator, whereas Japanese or Korean articles were translated by one of the authors (CHR). Reference lists of related articles were also searched. Conference abstracts were considered if they met the inclusion criteria and were exhibited in established conferences (e.g., including but not limited to ASCO, ASTRO, AASLD, EASL, ESTRO, ESMO, and RANZCR). All searching and inclusion processes were performed by two independent researchers, and any disagreement was resolved by mutual discussion and repetition of the search.

### 2.2. Data Items and Collection

The primary endpoint of the present study was grade ≥3 complications and the secondary endpoint was overall survival (OS). We used standardized sheets to collect the following data: (1) general information including names of authors, year of publication, affiliation, conflicts of interests, year of patient recruitment, number of patients; (2) clinical information including target disease, rate of Child–Pugh class A, EBRT modality and dose, the target of EBRT, sorafenib dose; (3) outcomes of interest including rates of grade ≥3 gastrointestinal, hepatologic, hematologic, and dermatologic complications, grade 4 or 5 complications, radiation-induced liver disease (RILD), and OS. Data extraction processes were performed by two independent researchers, and any disagreement was resolved by re-evaluation of the literature and discussion.

### 2.3. Risk of Bias and Assessment of the Quality

Because the application of concurrent sorafenib and EBRT has never been assessed in randomized trials, the possible risk of bias was discussed based on the Cochrane handbook chapter for the assessment of non-randomized studies [27]. The primary outcome of interest was treatment toxicity. Considering possible subjectivity in the assessment of treatment toxicities, and to facilitate comprehensive quantitative synthesis, we evaluated the possible toxicities by using four clinical categories (gastrointestinal, hepatologic, hematologic, and dermatologic). In addition, descriptions of complications in the studies were qualitatively analyzed. Regarding pooled analyses of OS, odds ratios between comparative arms (concurrent treatment vs. radiation therapy (RT) without sorafenib or sorafenib without RT) were synthesized, rather than pooled analyses of OS percentile rates, considering the diversity of target disease among studies. Because the searched studies were mostly observational studies, the Newcastle–Ottawa scale [28] was used for quality assessment.

### 2.4. Statistics

The principal summary measures were percentile rates of categorized grade ≥3 complications. OS was also pooled and analyzed, which yielded an odds ratio, using a median survival or 1-year OS rate. Median survival periods were estimated as mean values as necessary, using the method suggested by Hozo et al. [29]. The random effects model was used for pooled analyses of primary endpoints because the majority of studies were observational studies; this was in accordance with the recommendation in the Cochrane handbook that the random effects model should be the default model for analyzing non-randomized studies [30]. Heterogeneity assessment was performed using the Cochran Q [31] test and I^2^ statistics [32]; significant heterogeneity was considered to be present when a *p*-value of < 0.1 and I^2^ > 50% were obtained. Publication bias was assessed using visual inspection of funnel plots and the quantitative Egger’s test [33] for pooled analyses involving 10 studies. Possible publication bias was deemed to present with visual asymmetry in funnel plot analysis and a two-tailed *p*-value of < 0.1 in Egger’s test. Grade 4–5 complications, RILD, and OS were also qualitatively evaluated. Subgroup analyses were performed for studies focusing on intrahepatic versus non-intrahepatic lesions (e.g., denote targeting extrahepatic metastases or vessel involvement only), as local targets of EBRT, and a *p*-value below 0.1 denotes a statistically significant subgroup difference [34]. All statistical analyses were performed using Comprehensive Meta-Analysis version 3 (Biostat Inc., Englewood, NJ, USA).

### 2.5. Protocol Registration

This study is registered in PROSPERO (ID: CRD42021248705).

## 3. Results

### 3.1. Study Selection and Characteristics

Among the 961 initially identified studies, we excluded 586 studies owing to irrelevant formats and 22 studies owing to duplication among databases. The abstract screening was performed on 295 studies, and a full-text review was performed on 47 studies after excluding 248 studies with irrelevant subjects or formats. A full-text review was performed to identify studies that fully met the inclusion criteria, and 11 studies were finally included in the present study [16,18,19,20,21,35,36,37,38,39,40]. The selection process is illustrated in Figure 1.

Among the included studies, eight were from China or Taiwan, and one each was from Canada, Japan, and Korea. Nine studies were published as full-text articles, and two were abstracts published by the American Society for Radiation Oncology (ASTRO). All are written in English except one study written in Chinese [35]. The earliest study recruited patients from 2006 to 2009 and the latest from 2007 to 2017. Six studies were designed to have comparative arms (e.g., those with patients who underwent concurrent treatment vs. treatments involving EBRT without sorafenib or sorafenib without EBRT), and five were single-arm studies involving patients who received concurrent treatment. Rates of Child–Pugh A ranged from 65.6% to 100% (median 100%). Regarding RT modality, IMRT or tomotherapy was performed in five studies, SBRT in two studies, conventional 3-dimensional conformal RT in two studies, and brachytherapy in two studies. The majority of studies prescribed sorafenib in doses of 400 mg bid and were modified considering toxicities. The general characteristics of the included studies are summarized in Table 1.

### 3.2. Quality Assessment and Selection of Studies

Regarding quality assessment using NOS [28], four queries in the selection category were mostly fulfilled by all studies; because patients’ characteristics were representative of the population, outcomes of interest (e.g., complications or survival) were not present at the beginning of the study, and treatment modalities were well defined. Regarding queries in the outcome category, follow-up was adequate in the majority of studies included because they evaluated survival or toxicity for a sufficient period in patients with relatively short expected survival. Regarding comparability, we allotted two points (full point) for studies involving comparative arms without significant differences in clinical characteristics, or those that used propensity matching methods. These studies were considered to have reliable comparability. One point was allotted to studies involving comparative arms, but without statistical comparison. Single-arm studies were allotted zero points. Overall, five comparative studies were allotted nine points, one comparative study was allotted eight points, and five single-arm studies were allotted seven points. The detailed results are shown in Appendix A.

### 3.3. Summary of Individual Study Results

The rate of grade ≥3 gastrointestinal, hepatologic, hematologic, and dermatologic complication ranged from 0% to 20%, 0% to 20%, 0% to 33.3%, and 0% to 16.7%, respectively. The median value of the median survival period of patients who underwent concurrent application of sorafenib and RT was 17.4 months (range: 7.8–31.2). Among six studies with comparative design [20,21,35,38,39,40], none of the studies reported significant differences in grade ≥3 toxicity between arms, in quantitative and qualitative terms. Regarding OS comparison, four of six studies [20,21,35,39] reported that the OS benefit of concurrent treatment was significantly higher than that of non-concurrent treatment; one study reported a non-significant trend [40], and one study reported no difference [38]. The clinical results of the studies are shown in Table 2.

### 3.4. Synthesized Results and Qualitative Analyses of Endpoints

Pooled analyses of grade ≥3 toxicities were performed categorically: gastrointestinal (e.g., duodenal or gastric ulcer and/or perforation, abdominal pain, severe nausea and/or vomiting), hepatologic (e.g., elevation of liver function, symptoms of liver decompensation), hematologic (e.g., thrombocytopenia, leukopenia, anemia), and dermatologic complications were measured. Pooled rates of gastrointestinal, hepatologic, hematologic, and dermatologic grade ≥3 toxicities were 8.1% (95% confidence interval (CI): 4.8–13.5; heterogeneity: *p =* 0.444, I^2^ = ~0%), 12.9% (95% CI: 7.1–22.1, *p =* 0.259, I^2^ = 22.4%), 9.1% (95% CI: 3.8–20.3, *p =* 0.045, I^2^ = 51.3%), and 6.8% (95% CI: 3.8–11.7, *p =* 0.619, I^2^ = ~0%), respectively.

In subgroup analyses, pooled grade≥3 hepatologic toxicity rates were significantly lower in studies targeting non-intrahepatic than intrahepatic lesions (3.3% vs. 17.1%, *p =* 0.041), as targets of EBRT, and pooled hematologic toxicity rates showed a similar trend (3.3% vs. 16.0%, *p =* 0.078). Pooled rates of gastrointestinal and dermatologic grade ≥3 complications were not significantly different between the subgroups. Pooled analyses regarding OS were performed in six comparative studies, and concurrent treatment was more beneficial than non-concurrent treatment (odds ratio: 3.3, 95% CI: 1.3–8.59, p for odds ratio: 0.015; heterogeneity *p =* 0.002, I^2^ = 73.5).

The heterogeneity of pooled analyses was non-significant except for the pooled analyses of OS and hematologic toxicities in all studies and studies targeting intrahepatic lesions. Publication bias assessment was not performed, as all analyses included fewer than 10 studies. The pooled results are shown in Table 3 and are depicted in Figure 2 as forest plots.

### 3.5. Qualitative Assessment of Grade 4 or 5 and Descriptive Toxicities

Among nine studies with available data, four studies [16,18,19,39] reported few cases of grade 4 or 5 toxicities, whereas other studies did not. Brade et al. [16] reported lethal toxicity caused by upper gastrointestinal bleeding and tumor rupture. Among six comparative studies, two studies reported higher incidence of grade 1 or 2 hematologic or dermatologic toxicities in combination arms [21,40]. Kang et al. [35]. reported that grade 1 or 2 hematologic toxicity, fatigue and nausea are not different between comparative arms. Liu et al. [38]. also reported that toxicity was not different regarding radiation-induced liver toxicity and gastrointestinal bleeding. On the other hand, Zhang et al. [39] reported that additional brachytherapy showed benefits in terms of symptoms related to portal hypertension. The qualitative interpretation of toxicities is shown in detail in Table 4.

## 4. Discussion

Our findings suggest that concurrent application of sorafenib and EBRT is a feasible option for treating advanced HCC. Several types of adverse events of grade ≥3 occurred in 6%–13% of cases. Regarding the pooled analysis of OS, the combination therapy showed significantly greater benefit than did the non-concurrent therapy. On the basis of the above results, this study suggests that the application of concurrent therapy should be considered for advanced stages, in which available treatment options are limited.

The magnitude of toxicities and correlating recommendations for concurrent treatment vary among researchers. Brade et al. [16] from the Princess Margarette Hospital reported that the rate of grade ≥3 hepatotoxicity and gastrointestinal toxicity was approximately 20%, and hematotoxicity was as high as 33.3%. They did not recommend concurrent therapy because of the high risks of toxicities, assuming that toxicity was higher than that in their prior SBRT study with a similar design regarding RT modality. Chen et al. [18] reported a 22.5% hepatotoxicity of grade ≥3 and did not recommend the application of combination therapy. These studies have been published in reputed journals in the field of radiation oncology and have influenced many clinical decisions and research study designs.

In contrast, other studies included in the present meta-analysis were more favorable for the application of combination therapy. In terms of toxicities, comparative studies did not report excessive serious toxicities related to the concurrent application [20,21,35,38,39,40], and single-arm studies reported that the degree of complication was acceptable and severe toxicity (grade ≥4) was rare [19,36,37]. Furthermore, Zhang et al. [39] reported that the concurrent application of brachytherapy to the portal vein significantly reduced symptoms due to liver compensation. Five of six comparative studies [20,21,35,39,40] reported that combination therapy could increase OS. Four of these five studies [21,35,39,40] had reliable comparability, which means that the clinical characteristics were comparable between groups, or the concurrent treatment arm did not have relatively favorable clinical indicators. Theoretically, spatial cooperation in which EBRT targets macroscopic disease and sorafenib controls latent microscopic disease can be the basis for combination therapy [8]. Furthermore, sorafenib inhibits DNA damage repair in cancer cells in the tumor microenvironment and enhances the oxygen effect through normalization of the surviving tumor vasculature [14,15]. Sorafenib targets multiple targets (Vascular endothelial growth factor, Platelet-derived growth factor receptor, and Raf), reducing tumor–stromal interactions related to carcinogenesis and metastatic capacity, and affects metabolic capacity related to tumorigenicity [41,42], whereas EBRT induces mitotic death via DNA damage as well as immunogenic stimulation [43]. From a practical perspective, delaying the application of systemic agents during the 1–2-month period required for the planning and conducting of EBRT might lead to the development of latent microscopic disease while treating advanced HCCs. Therefore, the concurrent use of sorafenib and EBRT might be justified based on the above clinical results and preclinical rationale. Our results might also support the conduction of clinical trials, for instance, in which concurrent treatment is applied to patients with predictive factors related to poor outcomes after sorafenib [44].

Notably, subgroup analyses showed that rates of grade ≥3 hepatotoxicities and gastrointestinal toxicities were significantly different; these were lower in study groups targeting non-intrahepatic lesions than intrahepatic lesions. Currently, it might be difficult to recommend the application of concurrent treatment for all advanced HCCs because the literature regarding the feasibility and efficacy of concurrent treatment is insufficient, and some studies have reported risks of possible excessive toxicities [16,18]. However, considering the results of the present meta-analysis and the preclinical rationale mentioned above, we recommend the application of combination therapy to treat metastatic lesions or vascular tumor involvement as a target of EBRT. Combination therapy in such a disease situation is less likely to cause excessive toxicities, especially hepatotoxicity, which is a major concern. Notably, two studies reported a significant risk of hepatotoxicity, and the median value of the largest tumor size was 8.2–8.7 cm [16,18]. Targeting such large intrahepatic tumors inevitably leaves less normal liver volume unirradiated, which might result in excessive hepatotoxicity. Thus, concurrent treatment should be administered cautiously, considering tumor size and liver function reserve. In addition, skin reactions should be cautiously monitored when concurrent treatment is applied because excessive dermatologic toxicity has been reported in other case series [17], and, among the included studies, two of them [21,40] reported that grade 1 or 2 skin reactions were increased in addition to concurrent modality. Future studies are warranted to identify the mechanism of possible excessive toxicities caused by the concurrent application of sorafenib and EBRT.

The limitations of this study are as follows. Meta-analysis of observational studies is controversial because differences in study design and clinical characteristics might affect pooled estimates [45]. However, in the field of oncology, not all clinical decisions can be made on the basis of data from randomized studies. For intractable disease situations where standard treatment is not established and related literature is scarce, a meta-analysis of observational studies could be one of the few available options to suggest therapeutic decisions [46,47]. Efforts to improve the quality of meta-analysis, such as heterogeneity analysis, formal quality assessment, and sensitivity analyses, are recommended [46,48]. To optimize the present meta-analysis, we conducted statistical complements, as well as qualitative assessments. The heterogeneity in the pooled analyses was mostly low, suggesting that these results were reliable for clinical decisions. The inclusion of a small number of studies is also a limitation. We hope that this study would encourage further research to accumulate clinical evidence for navigating optimal indications of concurrent EBRT and sorafenib.

## 5. Conclusions

The present meta-analysis of the literature generates the clinical hypothesis that concurrent treatment of advanced HCC with sorafenib and EBRT could be a viable option. Such a modality can be applied to target metastatic lesions or vessel tumor involvement with RT. Treatment of intrahepatic lesions with a combined treatment modality should be performed cautiously by considering the target size and hepatic function reserve. Considering that concurrent treatment can increase treatment efficiency through spatial cooperation or radiosensitization and help avoid delays in systemic treatment, the clinical application should be more openly considered, and more future studies should be conducted.

## Figures and Tables

**Figure 1 cancers-13-02912-f001:**
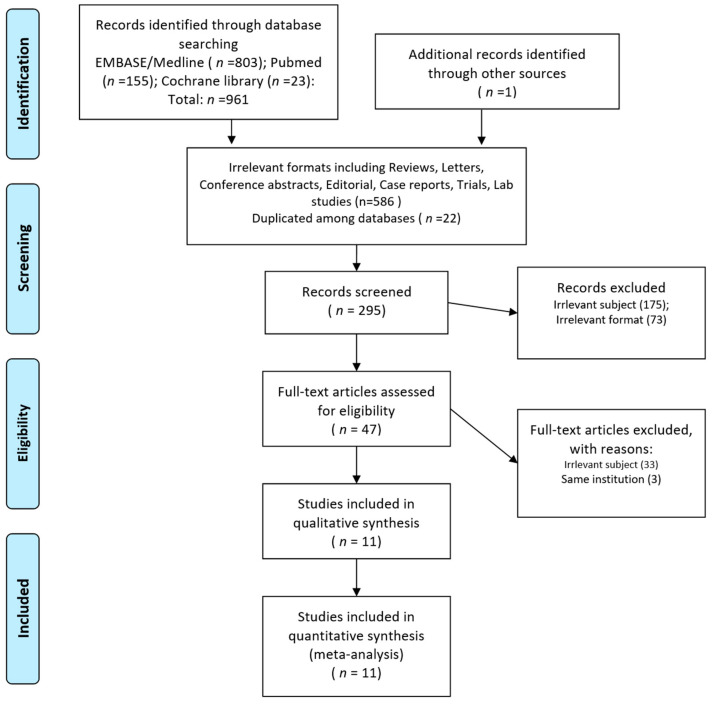
Study inclusion process.

**Figure 2 cancers-13-02912-f002:**
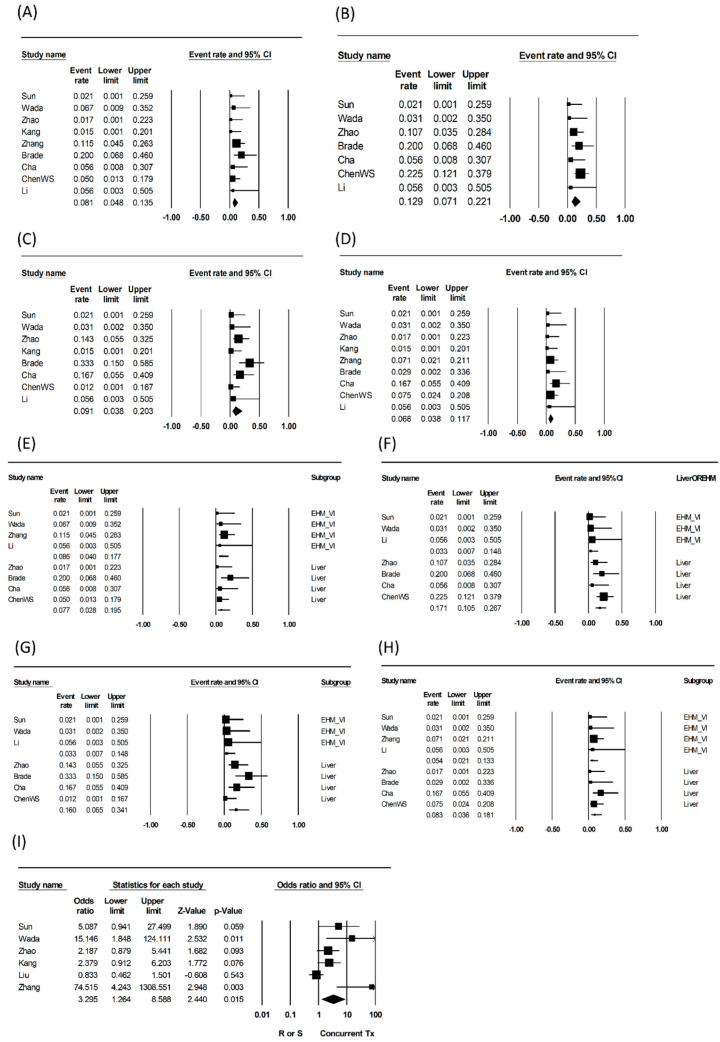
Forest plots of pooled analyses of all included studies regarding (**A**) gastrointestinal, (**B**) hepatologic, (**C**) hematologic, (**D**) dermatologic grade ≥3 toxicities; subgroup analyses comparing studies with RT targets of intrahepatic and non-intrahepatic lesions, regarding (**E**) gastrointestinal, (**F**) hepatologic, (**G**) hematologic, (**H**) dermatologic grade ≥3 toxicities. (**I**) Overall survival comparing concurrent and non-concurrent arms.

**Table 1 cancers-13-02912-t001:** Study characteristics.

Author	Publication, Year	Affiliation	Recruitment, Year	Study Design	Target Disease	No. of Patients *	CPC A (%)	Sorafenib	RT Modality	Dose	Other Combination Tx.
Sun [20]	Oncotarget, 2016	Fudan Univ, China	2011–2014	Comparative	Lung metastases	RS 23 R 22	98%	400 mg bid	Tomotherapy	50 Gy/5F or 10 F	
Wada [21]	Inters Med, 2018	Kyushu Medical Center, Japan	2009–2015	Comparative	Advanced HCC (MVI, EHM)	RS 15 S 47	100%	starting 800 mg/day	3DCRT	M50 (30–60)	
Zhao [40]	Frontier Oncol, 2019	Peking University, China	2015–2018	Comparative	HCC with MVI	TRS 28 TR 35	100%	400 mg bid	IMRT	5054 Gy, conventional fraction	TACE
Kang [35]	Chin J Clin Oncol, 2013	Navy General Hospital, China	2007–2009	Comparative	Recurrent and metastatic HCC	RSh 32 Rh 39	64.8%	400 mg bid	Gammaknife SBRT	36–50 Gy, 10–13 fractions	Hyperthermia
Liu [38]	ASTRO abstract, 2020	Chang Gung Memorial hospital, Taiwan	2007–2017	Comparative	Locally advanced HCC	RS 73 R 73, propensity matched					
Zhang [39]	World J Gastroenterol, 2017	Fudan Univ, China	2009–2015	Comparative	HCC MPVTT	TRS 37 TS 31	86.8%	400 mg bid	BrachyTx (I^125^)	mean accumulated dose 62.9 +/−2.3 Gy	TACE
Brade [16]	Int J Radiat Biol Phys, 2016	Princess Margaret Hospital, Canada	2009–2012	Single arm	HCC not amenable for other local Tx (PVT 63%)	15	100	200 mg OD-400 mg BID	SBRT	30–51 Gy/6 F	
Cha [19]	Yonsei Med J, 2013	Yonsei Cancer Center, Korea	2007–2011	Single arm	Liver HCC	13	85	400 mg bid (92%) 200 mg bid (8%)	3D CRT (92%)	M4 45 (30–54) in 1.8–5 Gy/F	
Cha-2 [19]					EHM	5		400 mg bid (60%) 200 mg bid (40%)	Tomotherapy	M50.4 (30–58.42)	
Chen B [36]	ASTRO abstract, 2019	Peking University, China	2010–2016	Single arm	HCC with MVI	8	100		IMRT	M50 (28–66)	
Chen WS [18]	Int J Radiat Biol Phys, 2014	3 hospitals in Taiwan	2010–2013	Single arm	Locally advanced HCC	40	100%	starting 400 mg bid	IMRT	50–60 Gy in 2–2.5 Gy/F	
Li [37]	J Cancer Res Clin Oncol, 2010	Sun Yat-Sen Univ, Taiwan	2006–2009	Single arm	Lung metastases	8		400 mg bid	BrachyTx (I^125^)	Minimial peripheral dose 120–160 Gy	

Abbreviations: HCC, hepatocellular carcinoma; MVI, major vascular invasion; EHM, extrahepatic metatases; 3DCRT, 3-dimensional conformal radiotherapy; IMRT, intensity-modulated radiotherapy; HBV, hepatitis B virus; MPVTT, main portal vein tumor thrombosis; brachyTx., brachytherapy * RS, radiotherapy and sorafenib; TRS, TACE and radiotherapy and sorafenib; RSh, radiotherapy and sorafenib and hyperthermia.

**Table 2 cancers-13-02912-t002:** Clinical results of studies included.

Author	RT target	No. of Patients *	Reliable Comparability	OS	Grade ≥3 Toxicity: GI	Hepatologic	Hematologic	Dermatologic
Sun	EHM (lung mets)	RS 23 R 22	No	RS: 91.1% (1 y), 78.8% (2 y) R: 66.8% (1 y), 30.4% (2 y) (*p =* 0.007)	RS, R: 0%	RS, R: 0%	RS, R: 0%	RS, R: 0%
Wada	EHM or MVI (not intrahepatic)	RS 15 S 47	Yes	RS: M31.2mo., 93.3% (1 y), 56.9% (2 y) S: M12.1mo., 47.9% (1 y), 12.4% (2 y); (*p* < 0.01)	RS: 6.7% S: 2.2% (*p =* NS)	RS: 0% S: 6.4% (*p =* 0.23)	RS: 0% S: 0%	RS: 0% S: 4.3% (*p =* 0.24)
Zhao	Liver HCC	TRS 28 TR 35	Yes	TRS: M19 mo TR: M15.2 mo (*p =* 0.094)	TRS: 0% TR: 0%	TRS: 10.7% TR: 11.4% (*p =* 1.0)	TRS: 14.3% TR: 17.1% (*p =* 1.0)	TRS: 0% TR: 0%
Kang	EHM or Liver HCC	RSh 32 Rh 39	Yes	RSh: 62.5% (1 y) Rh: 41.2% (1 y) (*p =* 0.048)	RSh, Rh: 0%	Not assessable	RSh, Rh: 0%	RSh, Rh: 0%
Liu	Liver HCC	RS 73 R 73, propensity matched	Yes	RS: M9.6 R: M9.9 (*p =* 0.544)				
Zhang	PVT only	TRS 37 TS 31	Yes	TRS: 54.3% (1 y), 14.1% (2 y) TS: 0% (1 y) (*p* < 0.001)	TRS: 11.5% (diarrhea) TS: 3.6% (diarrhea) (*p =* NS)	Grade not assessed		TRS: 7.1% (HFS) TS: 3.6% (HFS) (*p =* NS)
Brade	Liver HCC	15	NA	M26.3 mo, 62.5% (1 y)	20% (GI bleeding and SBO)	20% (LFT elevation)	33.3% (thrombocytopenia)	0%
Cha	Liver HCC	13	NA	M7.8 mo, 35% (1 y)	5.6% (DU bleed)	5.6% (LFT elevation)	16.7% (thrombocytopenia)	16.7% (HFS)
Cha-2	EHM	5	NA	M15.7 mo, 60% (1 y)				
Chen B	Liver HCC	8	NA					
Chen WS	Liver HCC	40	NA	M14 mo, 52.5% (1 y), 32% (2 y)	5% (diarrhea)	22.50%	0%	7.5% (HFS)
Li	EHM (lung mets)	8	NA	M21 mo, 100% (1 y), 50% (2 y)	0	0	0	0

Abbreviations: RT, radiotherapy; OS, overall survival; RILD, radiation-induced liver toxicity; EHM, extrahepatic metastases; HCC, hepatocellular carcinoma; brachTx., brachytherapy * RS, radiotherapy and sorafenib; TRS, TACE and radiotherapy and sorafenib; RSh, radiotherapy and sorafenib and hyperthermia.

**Table 3 cancers-13-02912-t003:** Pooled rates of grade 3 or higher toxicities.

Subject Studies	No. of Studies	Patients Underwent Concurrent Treatment	Heterogeneity *p*	I^2^	Pooled Rate (95% CI)	Subgroup Comparison *p*
*Gastrointestinal toxicity*						
All studies	9	217	0.444	~0%	8.1% (4.8–13.5)	NA
Non–intrahepatic	4	83	0.649	~0%	8.6% (4.0–17.7)	0.859
Intrahepatic	4	102	0.198	35.8%	7.7% (2.8–19.5)	
*Hepatologic toxicity*						
All studies	7	148	0.259	22.4%	12.9% (7.1–22.1)	NA
Non-intrahepatic	3	46	0.882	~0%	3.3% (0.7–14.8)	0.041
Intrahepatic	4	102	0.366	5.4%	17.1% (10.5–26.7)	
*Hematologic toxicity*						
All studies	8	180	0.045	51.3%	9.1% (3.8–20.3)	NA
Non-intrahepatic	3	46	0.882	~0%	3.3% (0.7–14.8)	0.078
Intrahepatic	4	102	0.08	55.6%	16.0% (6.5–34.1)	
*Dermatologic toxicity*						
All studies	9	217	0.619	~0%	6.8% (3.8–11.7)	NA
Non-intrahepatic	4	83	0.839	~0%	5.4% (2.1–13.3)	0.485
Intrahepatic	4	102	0.321	14.3%	8.3% (3.6–18.1)	
*Overall survival*	6	455	0.002	73.5%	OR: 3.3 (1.3–8.59, *p =* 0.015)	

Abbreviations: CI, confidence interval; OR, odds ratio; NA, not assessable.

**Table 4 cancers-13-02912-t004:** Qualitative interpretation of toxicities.

Author	RT Target	No. of Patients *	Grade 4 or 5 Toxicities RILD	Qualitative Interpretation
Sun	EHM (lung mets)	RS 23 R 22	0% no RILD	All toxicities were G1 or 2 toxicities
Wada	EHM or MVI (not intrahepatic)	RS 15 S 47		Grade 1 or 2 hematologic, dermatologic adverse events were higher in the RS group Overall grade ≥3 toxicity incidences are similar (20% vs. 19.2%, *p* = NS)
Zhao	Liver HCC	TRS 28 TR 35	0% no RILD	All skin reactions and HFS were G1 or 2 toxicities, but these toxicities were of a higher grade with TRS (92.9% & 17.9% vs. 68.6% & 0%)
Kang	EHM or Liver HCC	RSh 32 Rh 39	0% RILD not assessed	No significant difference in G1 or 2 BM suppression, fatigue, nausea between arms Overall G3 complication 9.4%
Liu	Liver HCC	RS 73 R 73, propensity matched		No significant difference in RILD and GI bleeding
Zhang	PVT only	TRS 37 TS 31	1 case of HTN G4 (3.2%) in TRS no lethal toxicity	Adding brachyTx. improved portal hypertension symptoms (new ascites, liver dysfunction), and OS (*p* < 0.001, 1 yr OS 54.3% vs. 0%)
Brade	Liver HCC	15	1 case of liver enzyme change G4; 1 case of SBO G4 1 case of upper GI bleeding, rupture and death (6.7%)	
Cha	Liver or EHM	18	1 case of G4 thrombocytopenia (5.6%) no RILD	
Chen B	Liver HCC	8	No lethal toxicity no RILD	
Chen WS	Liver HCC	40	4 cases (11.1%) G4-5 hepatic toxicity RILD 15% (6 cases, 3 of which died without tumor progression)	
Li	EHM (lung mets)	8	No RILD no lethal toxicity	

Abbreviations: RT, radiotherapy; RILD, radiation-induced liver disease; EHM, extrahepatic metastases; MVI, major vessel invasion; HCC, hepatocellular carcinoma; BM, bone marrows; HFS, hand-foot syndrome; SBO, small bowel obstruction; GI, gastrointestinal * RS, radiotherapy and sorafenib; TRS, TACE and radiotherapy and sorafenib; RSh, radiotherapy and sorafenib and hyperthermia.

## Data Availability

The authors confirm that the data supporting the findings of this study are available within the article and/or its Appendix A.

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
