# Peer review of "Is the Concurrent Use of Sorafenib and External Radiotherapy Feasible for Advanced Hepatocellular Carcinoma? A Meta-Analysis"

_cancers, 2021, doi:10.3390/cancers13122912_

Round 1

Reviewer 1 Report

Here are my suggestions/comments to the manuscript untitled:” Is the concurrent use of sorafenib and external radiotherapy feasible for advanced hepatocellular carcinoma? A meta-analysis” written by Rim et al. and submitted to Cancers journal.

  • Short title should be modified to enhance clarity – by reading it we should understand what the displayed study is talking about, here it is not.
  • In the Introduction part, a paragraph explaining why HCC is lately diagnosed should definitely be added – indeed, it would then become logic to talk about “advanced HCC” as mentioned all over the manuscript, and the combined use of sorafenib and radiotherapy. To do so, please refer and cite these 4 key papers:
    • Ayuso C, Rimola J, García-Criado A. Imaging of HCC. Abdom Imaging. 2012 Apr;37(2):215-30. doi: 10.1007/s00261-011-9794-x. PMID: 21909721.

    • Chalasani N, Horlander JC Sr, Said A, Hoen H, Kopecky KK, Stockberger SM Jr, Manam R, Kwo PY, Lumeng L. Screening for hepatocellular carcinoma in patients with advanced cirrhosis. Am J Gastroenterol. 1999 Oct;94(10):2988-93. doi: 10.1111/j.1572-0241.1999.01448.x. PMID: 10520857.

    • Burrel M, Llovet JM, Ayuso C, Iglesias C, Sala M, Miquel R, Caralt T, Ayuso JR, Solé M, Sanchez M, Brú C, Bruix J; Barcelona Clínic Liver Cancer Group. MRI angiography is superior to helical CT for detection of HCC prior to liver transplantation: an explant correlation. Hepatology. 2003 Oct;38(4):1034-42. doi: 10.1053/jhep.2003.50409. PMID: 14512891.

    • Cassim S, Raymond VA, Lacoste B, Lapierre P, Bilodeau M. Metabolite profiling identifies a signature of tumorigenicity in hepatocellular carcinoma. Oncotarget. 2018 Jun 1;9(42):26868-26883. doi: 10.18632/oncotarget.25525. PMID: 29928490; PMCID: PMC6003570.

  • Still in the Introduction part, this sentence:” This meta-analysis aimed to improve the treatment strategy for advanced HCC and encourage the initiation of related 58 studies by evaluating the feasibility of such combination therapy.” should be attenuated. This meta-analysis aims to collect/compile published data in order to have a better understanding of whether using a combined treatment in advanced in HCC is feasible or not.
  • In the Discussion part, after:” Furthermore, sorafenib inhibits DNA damage repair in cancer cells in the tumor microenvironment and oxygenates tumor cells to increase radiation sensitivity [7, 8]”, the authors should explain in 1-2 sentences how HCC tumor microenvironment can impact on cell tumorigenicity and de facto immune cells, and most importantly how the metabolism of HCC cells is affected by such chemotherapy – it would emphasize the potential of using a combined therapy. To do so, please refer and cite these 2 papers:
    • Yang JD, Nakamura I, Roberts LR. The tumor microenvironment in hepatocellular carcinoma: current status and therapeutic targets. Semin Cancer Biol. 2011 Feb;21(1):35-43. doi: 10.1016/j.semcancer.2010.10.007. Epub 2010 Oct 12. PMID: 20946957; PMCID: PMC3050428.

    • Cassim S, Raymond VA, Dehbidi-Assadzadeh L, Lapierre P, Bilodeau M. Metabolic reprogramming enables hepatocarcinoma cells to efficiently adapt and survive to a nutrient-restricted microenvironment. Cell Cycle. 2018;17(7):903-916. doi: 10.1080/15384101.2018.1460023. Epub 2018 May 21. PMID: 29633904; PMCID: PMC6056217.

  • I appreciate that in the Discussion part, the authors included a paragraph dealing with the limitations of the study.

Author Response

Reviewer 1

Here are my suggestions/comments to the manuscript untitled:” Is the concurrent use of sorafenib and external radiotherapy feasible for advanced hepatocellular carcinoma? A meta-analysis” written by Rim et al. and submitted to Cancers journal.

Short title should be modified to enhance clarity – by reading it, we should understand what the displayed study is talking about, here it is not.

  • We appreciate and agree with your comment. We changed the title from “concurrent Sorafenib & EBRT” to “Meta-analysis on the feasibility of concurrent sorafenib & EBRT.”

In the Introduction part, a paragraph explaining why HCC is lately diagnosed should definitely be added – indeed, it would then become logic to talk about “advanced HCC” as mentioned all over the manuscript, and the combined use of sorafenib and radiotherapy. To do so, please refer and cite these 4 key papers:

Ayuso C, Rimola J, García-Criado A. Imaging of HCC. Abdom Imaging. 2012 Apr;37(2):215-30. doi: 10.1007/s00261-011-9794-x. PMID: 21909721.

 Chalasani N, Horlander JC Sr, Said A, Hoen H, Kopecky KK, Stockberger SM Jr, Manam R, Kwo PY, Lumeng L. Screening for hepatocellular carcinoma in patients with advanced cirrhosis. Am J Gastroenterol. 1999 Oct;94(10):2988-93. doi: 10.1111/j.1572-0241.1999.01448.x. PMID: 10520857.

 Burrel M, Llovet JM, Ayuso C, Iglesias C, Sala M, Miquel R, Caralt T, Ayuso JR, Solé M, Sanchez M, Brú C, Bruix J; Barcelona Clínic Liver Cancer Group. MRI angiography is superior to helical CT for detection of HCC prior to liver transplantation: an explant correlation. Hepatology. 2003 Oct;38(4):1034-42. doi: 10.1053/jhep.2003.50409. PMID: 14512891.

 Cassim S, Raymond VA, Lacoste B, Lapierre P, Bilodeau M. Metabolite profiling identifies a signature of tumorigenicity in hepatocellular carcinoma. Oncotarget. 2018 Jun 1;9(42):26868-26883. doi: 10.18632/oncotarget.25525. PMID: 29928490; PMCID: PMC6003570.

  • We appreciate and agree with your comment. We have described why HCC is commonly diagnosed at a late stage using the references you have suggested and some other studies. (first paragraph of the Introduction)

Still in the Introduction part, this sentence:” This meta-analysis aimed to improve the treatment strategy for advanced HCC and encourage the initiation of related studies by evaluating the feasibility of such combination therapy.” should be attenuated. This meta-analysis aims to collect/compile published data in order to have a better understanding of whether using a combined treatment in advanced in HCC is feasible or not.

  • We appreciate your thoughtful comment and suggestion. We have changed the last sentence of the Introduction to: “This meta-analysis aimed to compile published data to gain a better understanding of the feasibility of using a combined treatment in advanced HCC. ”

In the Discussion part, after:” Furthermore, sorafenib inhibits DNA damage repair in cancer cells in the tumor microenvironment and oxygenates tumor cells to increase radiation sensitivity [7, 8]”, the authors should explain in 1-2 sentences how HCC tumor microenvironment can impact on cell tumorigenicity and de facto immune cells, and most importantly how the metabolism of HCC cells is affected by such chemotherapy – it would emphasize the potential of using a combined therapy. To do so, please refer and cite these 2 papers:
Yang JD, Nakamura I, Roberts LR. The tumor microenvironment in hepatocellular carcinoma: current status and therapeutic targets. Semin Cancer Biol. 2011 Feb;21(1):35-43. doi: 10.1016/j.semcancer.2010.10.007. Epub 2010 Oct 12. PMID: 20946957; PMCID: PMC3050428.

Cassim S, Raymond VA, Dehbidi-Assadezadeh L, Lapierre P, Bilodeau M. Metabolic reprogramming enables hepatocarcinoma cells to efficiently adapt and survive to a nutrient-restricted microenvironment. Cell Cycle. 2018;17(7):903-916. doi: 10.1080/15384101.2018.1460023. Epub 2018 May 21. PMID: 29633904; PMCID: PMC6056217.

  • We appreciate and agree with your thoughtful comments. We added two sentences referencing the literature suggested and some other studies in the discussion about the relationship between the HCC microenvironment and concurrent treatment with sorafenib and radiotherapy (page 9, first paragraph). As we are clinicians and have less knowledge in molecular oncology, and considering that the main subject of the present study is clinical, we added a brief discussion that may not have sufficient detail. If there is anything further you would like us to adjust or add, your suggestions would be more than welcomed.

I appreciate that in the Discussion part, the authors included a paragraph dealing with the limitations of the study

  • We do appreciate your kind comment.

Reviewer 2 Report

Chai Hong Rim et al. report about the concurrent treatment of advanced HCC with sorafenib and EBRT, uncovering it to be considered and applied to target metastatic lesion or local vessel involvement. The manuscript is of interest.

Point to be considered:

  1. The authors themselves stated "Efforts to improve the quality of meta-analysis, such as heterogeneity analysis, formal quality assessment, and sensitivity analyses, are recommended".  Both false-positive and false-negative signals of heterogeneitynmay be anticipated based on some rational considerations. Simple suggestions of how to avoid these flaws might suffice the scope of the manuscript, nonetheless, given the above mentioned issue, I would suggest to tune-down the con conclusion and considering the manuscript as an hypotesis-generating meta-analysis.
  2. The rationale of why the authors came up with this manuscript should be better highlighted
  3. What is the information that is not exactly available that motivated the authors to come up with this information. What are the current caveats and how do the authors highlight the current research in answering them? If not they need to address in future directions.
  4. The authors need to highlight what new information the review is providing to enhance the research in progress. For instance, the authors mentioned the rational role of sorafenib synergism. This reviewer personally misses some insights regarding additional details regarding the predictive fators to sorafenibe response, i.e. unlike other targeted therapies, predictive and prognostic markers in HCC patients treated with sorafenib are lacking (refer to PMID: 31640191).

Author Response

Chai Hong Rim et al. report about the concurrent treatment of advanced HCC with sorafenib and EBRT, uncovering it to be considered and applied to target metastatic lesion or local vessel involvement. The manuscript is of interest.

Point to be considered:

1. The authors themselves stated "Efforts to improve the quality of meta-analysis, such as heterogeneity analysis, formal quality assessment, and sensitivity analyses, are recommended".  Both false-positive and false-negative signals of heterogeneity may be anticipated based on some rational considerations. Simple suggestions of how to avoid these flaws might suffice the scope of the manuscript, nonetheless, given the above mentioned issue, I would suggest to tune-down the conclusion and considering the manuscript as an hypothesis-generating meta-analysis.

  • We agree and appreciate your thoughtful comments. Therefore, we removed the first sentence (which was most conclusive sentence in the original manuscript) as follows: “Our study suggests that concurrent treatment of advanced HCC with sorafenib and EBRT is a viable option,” and changed to “The present meta-analysis of literature generates clinical hypothesis that concurrent treatment of advanced HCC with sorafenib and EBRT could be a viable option.”

2.The rationale of why the authors came up with this manuscript should be better highlighted

  • We agree and appreciate your comment. We added a paragraph in the Introduction explaining that advanced HCC is still commonly diagnosed despite increased efforts for surveillance (page 3, first paragraph). Furthermore, we have adjusted the last sentence of the Introduction to indicate the purpose of the current study. This meta-analysis aimed to improve the treatment strategy for advanced HCC and encourage the initiation of related studies by evaluating the feasibility of such combination therapy “This meta-analysis aimed to compile published data to gain a better understanding of whether using a combined treatment in advanced HCC is feasible.” Page 3, last paragraph:

3. What is the information that is not exactly available that motivated the authors to come up with this information. What are the current caveats and how do the authors highlight the current research in answering them? If not they need to address in future directions.

  • We appreciate your comment. We have added sentences indicating the current caveats and what is unknown to motivate the present study (Page 3, last paragraph, first and second sentences).

4. The authors need to highlight what new information the review is providing to enhance the research in progress. For instance, the authors mentioned the rational role of sorafenib synergism. This reviewer personally misses some insights regarding additional details regarding the predictive fators to sorafenibe response, i.e. unlike other targeted therapies, predictive and prognostic markers in HCC patients treated with sorafenib are lacking (refer to PMID: 31640191).

  • Thank you for your important comment. We hope that this study can stimulate future studies that apply concurrent sorafenib and RT for patients with predictive factors related to poor outcomes after sorafenib. Although sorafenib does not currently have enough predictive markers as compared with other target agents, clinical studies involving both sorafenib and radiotherapy might provide a new perspective on such subjects. Therefore, we have added a relevant discussion incorporating the suggested reference (page 9, end of the first paragraph).

Reviewer 3 Report

Hepatocellular carcinoma (HCC) is a poor prognosis tumor. Therefore, multimodality therapies are frequently used to treat the patients with advanced HCC. In this manuscript, the authors evaluated the feasibility of a concurrent application of sorafenib and external beam radiation therapy for the advanced HCC. Eleven studies involving 512 patients were included in this meta-analysis. The authors found that the pooled rates of gastrointestinal, hepatological, hematological, and dermatological grade ≥3 toxicities were 8.1%, 12.9%, 9.1%, and 6.8%, respectively, that the pooled grade ≥3 hepatological and hematological toxicity rates were lower in studies targeting non-intrahepatic lesions than those targeting intrahepatic lesions, and that, regarding the overall survival, concurrent treatment was more beneficial than non-concurrent treatment (odds ratio, 3.3). They also found that one study reported a case of lethal toxicity due to tumor rupture and gastrointestinal bleeding. So the authors concluded that concurrent treatment of advanced HCC with sorafenib and EBRT can be applied to target metastatic lesions or local vessel involvement, and that intrahepatic lesions should be treated cautiously by considering the target size and hepatic function reserve.

This is a meta-analysis on the concurrent treatment of sorafenib and external beam radiation therapy for advanced hepatocellular carcinoma. The data analysis was appropriate and the manuscript was well prepared. This article can provide useful information for the clinicians to manage the advanced HCC patients. 

Author Response

This is a meta-analysis on the concurrent treatment of sorafenib and external beam radiation therapy for advanced hepatocellular carcinoma. The data analysis was appropriate and the manuscript was well prepared. This article can provide useful information for the clinicians to manage the advanced HCC patients. 

  • We appreciate your kind and review. We also hope that our study can help in the management of patients with advanced HCC.

Round 2

Reviewer 1 Report

This is a nice and relevant study which deserves to be published. 

Reviewer 2 Report

The authors have clarified several of the questions I raised in my previous review. Most of the major problems have been addressed by this revision.